# Tuning the Sensitivity and Dynamic Range of Optical Oxygen Sensing Films by Blending Various Polymer Matrices

**DOI:** 10.3390/bios12010005

**Published:** 2021-12-22

**Authors:** Kaiheng Zhang, Siyuan Lu, Zhe Qu, Xue Feng

**Affiliations:** 1Part Rolling Key Laboratory of Zhejiang Province, School of Mechanical Engineering and Mechanics, Ningbo University, Ningbo 315211, China; geerheng086@gmail.com; 2Institute of Flexible Electronics Technology of THU, Jiaxing 314000, China; quzhe_2013@foxmail.com; 3Department of Engineering Mechanics, Tsinghua University, Beijing 100084, China; fengxue@tsinghua.edu.cn

**Keywords:** optical oxygen sensing film, phase delay method, microporous filter membrane, blending, sensitivity, dynamic range

## Abstract

In this work, eight different types of optical oxygen sensing films were prepared by impregnating indicator and matrix solution on the surface of a polypropylene microporous filter membrane. The polymer matrix of the sensing films was ethyl cellulose (EC), polymethyl methacrylate (PMMA), and their blends with different mixing ratios. Scanning electron microscopy (SEM), laser confocal microscopy, and fluorescence spectrometer were used to investigate the morphologies and optical properties of the sensing films. Phase delay measurements under different oxygen partial pressures (*P*_O__2_) and temperatures were applied to investigate the analytical performances of the sensing film for gaseous O_2_ monitoring. Results show that the response time of all the sensing films was extremely fast. The sensitivities and dynamic ranges of the sensing films with the blended polymer matrix were separately decreased and increased as the EC/PMMA ratio decreased, and the S-V curve of the sensing films blended with equal content of EC and PMMA exhibited good linearity under different temperatures, showing a promising prospect in practical application.

## 1. Introduction

Precise and fast sensing of the oxygen concentration in different environments, such as in blood, respiration gas, and water, has attracted particular attention in various fields, including environmental detection, medical monitoring, and industrial gas monitoring [1,2]. To this end, oxygen sensors (OSs) are becoming a potent device for biological and industrial oxygen measurement. The main methods for oxygen measurements can be categorized into three groups, based on their detecting principle: (i) the Winkler titration method [3], (ii) electrochemical analysis [4], and (iii) optical analysis [5]. The Winkler titration method cannot be practically applied due to its low detection efficiency [6]. The electrochemical oxygen sensor reflects the oxygen concentration via the magnitude of the current signal, with a sensitivity of 0.08–0.23, and a dynamic range from 0 to 21% (oxygen concentration) [7]. However, oxygen is consumed during the electrochemical process, which hinders the application of electrochemical methods in continuous oxygen measurement in a closed environment. Among all the methods, optical oxygen sensors based on the fluorescence quenching effect, which is induced by the collision between triplet oxygen and luminescent indicators, does not consume oxygen nor transmit electrical signals due to the particularity of the photophysical process during detection and exhibits a higher sensitivity (0.013–263), a wider dynamic range (0–100%, oxygen concentration) [6], a faster response [8], and immunity to external magnetic interference [9,10].

Generally, a typical optical oxygen sensor (OS) incorporates an optical signal detection device (OSD) and a sensing film. As the functional part, sensing film is mainly composed of fluorescent indicators and a matrix. Transition metal complexes, especially metalloporphyrin, such as Pt (II) complexes [11,12], are often used as indicators due to their long fluorescence lifetime, intense phosphorescence, and large Stokes’ shift (about 100–200 nm) [13,14,15]. These features are suitable for phase modulation systems and enable the separation of the excitation light signal from the fluorescence signals via a detection device. Polymers such as polystyrene (PS) [16], polydimethylsiloxane (PDMS) [17], ethyl cellulose (EC) [18], and polymethyl methacrylate (PMMA) [19], are widely used as the matrix of the sensing film. In addition, several researchers have reported that the combination of conjugated polymers (i.e., polythiophene, polyaniline, and polypyrene) with solid oxide electrolytes (i.e., TiO_2_ and SnO_2_) can greatly improve the detecting performance of gaseous sensors [20,21,22]. Normally, different types of polymers can affect the sensitivity of the sensing film due to the different oxygen permeability caused by varying degrees of void volume [23]. Under the same oxygen partial pressure, a polymer with higher oxygen permeability allows more oxygen molecules to contact the indicator, increases the quenching degree of the indicator, and results in higher sensitivity and a shorter dynamic range of the sensing film [24]. In contrast, the polymer with lower oxygen permeability permits less contact between oxygen molecules and the indicator in sensing film, resulting in lower sensitivity and a wider dynamic range. Thus, the sensing film made by higher oxygen permeability (i.e., EC) is suitable for low-concentration oxygen monitoring, while polymers with lower oxygen permeability (e.g., PMMA) are often used as the matrix for hyperbaric oxygen sensors [25].

Since the indicator is often mixed with the polymer matrix, a film that hosts the mixture of indicator and polymer also acts as the sensing film substrate. The structure of the film can dramatically affect the response time of the oxygen sensor. Compared with flat structures, the large specific surface area of the porous structure can improve [26] the collision efficiency between gas molecules and luminescent indicators, thus shortening the response time. Mao et al. [27] prepared a sensing film with honeycomb structured films by creating pores through the imprint of water droplets, which improved the sensitivity and responsivity of the oxygen sensor. Guo et al. [9] developed an oxygen sensor based on a porous plastic probe, which exhibited a fast response time (<90 s) to dissolved oxygen. However, the preparation of such porous films is time-consuming and needs strict control of the temperature, humidity, and viscosity of the evaporation solution. In addition, water or moisture may hamper the contact between oxygen and the indicator, and the increase in temperature can aggravate the collision quenching between indicators and oxygen molecules or change the permeability of the polymer matrix of the sensing films, which could jointly affect the detecting accuracy of oxygen sensors in the actual application. Therefore, the polypropylene microporous membrane, which exhibits a porous structure [28], high thermal stability, good hydrophobicity [29,30], and better chemical stability, is a promising substrate for sensing films.

In short, the difference in the intrinsic permeability of polymers towards oxygen hinders the fabrication of oxygen sensors that can adapt to both low-concentration and hyperbaric oxygen environments. Moreover, tuning the sensitivity by altering the polymer types also demands great intervention for a specific sensor system, since the matrix of the sensing film can only be used in a specific environment. Koren et al. [31] achieved a fine-tuning of the sensitivity and dynamic range of the optical oxygen sensors by changing different polystyrene derivatives or copolymers of two different styrene-derivative monomers as matrix materials. The underlying mechanism for such tuning was ascribed to the oxygen permeability difference of different polystyrene derivatives. Therefore, blending different types of polymers with disparate oxygen permeability as a matrix and utilizing a ready porous structural film could be a cost-effective approach to tune the sensitivity and dynamic range of optical oxygen-sensing films for specific environments. For example, the sensing film with a high sensitivity and low dynamic range is suitable for the production of respirators and air quality detectors, while the sensing film with low sensitivity and a high dynamic range is suitable for the production of hyperbaric oxygen chambers.

In the present work, a series of sensing films are produced by embedding the Pt (II) octaethylporphine (PtOEP) into a polymer matrix with different blending ratios of EC and PMMA on a polypropylene microporous membrane. The optical properties and surface morphologies of the sensing films with different blending ratios were investigated by a fluorescent spectrometer, scanning electron microscopy (SEM), and laser confocal microscopy. Phase delay methods were employed to test the sensitivities and response time of the sensing films with different blending ratios of EC and PMMA. Consequently, the performance parameters of different polymer-sensing films were measured. Our finding shows that the blending of different types of polymers can significantly tune the sensitivity and dynamic ranges of the sensing film, thus providing a new path for the preparation of the sensing films that adopt both low-concentration and hyperbaric oxygen environments. The sensing films prepared in this work display a wide dynamic range and sufficient sensitivities, showing a promising application potential in the field as respirators, hyperbaric oxygen chambers, air quality detectors, and hazardous gas alarms.

## 2. Theory

### 2.1. Stern-Volmer Relationship

Normally, it is assumed that the indicators are uniformly distributed in the polymer matrix and the microenvironment around each luminary remains the same. Then, the bimolecular collision process between oxygen and indicators can be described by the Stern-Volmer equation [32,33]:(1)$$\tau_{0}/{\tau =1+K}_{\mathrm{sv}}{\cdot P}_{O2}$$
where *τ*_0_ is the lifetime of indicators in the environment without oxygen, *τ* is the lifetime under a different oxygen partial pressure, *K*_SV_ is the Stern-Volmer quenching constant, and *P*_O__2_ is the oxygen partial pressure. Ideally, the Stern-Volmer plot (SVP) is linear, and *K*_SV_ is equal to the slope of SVPs, which visually represents the sensitivity of the sensing system.

However, since the media for the quenching processes is not always homogeneous, the standard quenching model based on ideal SVPs often fails to perfectly accommodate the practical conditions [34,35]. It is necessary to set up the quenching mechanism and derive the quenching equation based on the practical results.

### 2.2. Phase Delay Measurements

Although the oxygen partial pressure can be acquired by fluorescence intensity measurements, several limitations, such as background light source interference and laser light source intensity drift, will interfere with the accuracy of the relative results [36]. An alternative is to measure the fluorescence lifetime of the indicators under different oxygen partial pressures, which can be achieved from the time-domain or frequency-domain methods. For frequency-domain analysis, the lifetime of the indicator is determined by the delay phase between the phase angle of sinusoidally modulated excitation light and the fluorescence emitted from the indicator. The relationship between delay phase and lifetime can be described as [32]:(2)$$\tau_{0}/{\tau =\tan\theta }_{0}/\tan\theta$$
where *θ*_0_ is the delay phase obtained in the absence of oxygen and *θ* is the delay phase under different oxygen partial pressure.

## 3. Experiments

### 3.1. Chemical Reagents and Materials

Pt (II) octaethylporphine (PtOEP, purity 95%) was purchased from Sigma-Aldrich (Shanghai, China). Polymers including ethyl cellulose (EC, powder) and polymethyl methacrylate (PMMA, average M_w_ ~120,000 by GPC) were commercially available from Macklin Biochemical Co., Ltd. (Shanghai, China). The polypropylene microporous membrane was provided by Yibo Filter Equipment Factory (Haining, China). Chloroform was purchased from Aladdin Ltd. (Shanghai, China). All chemicals and reagents were stored and used as required.

### 3.2. Synthesis of the Sensing Films

Prior to the preparation of the sensing films, the microporous filtration membrane was rinsed via deionized water and ethanol in sequence to eliminate the contaminants, then dried in a vacuum drying chamber for 1 h. Then, 4 mg of PtOEP was mixed with 80 mg of polymer powder in 4 mL CHCl_3_ by ultrasonic oscillation for 1 h to form a uniform fluorescent solution. EC, PMMA, and their blending with different weight ratios (e.g., EC/PMMA ratio for 1:1, 1:3, and 3:1) were chosen for the polymer matrix of the sensing films. Afterward, the microporous filtration membrane was impregnated into the fluorescent solution for 1–2 s and then dried at room temperature to evaporate the solvent (CHCl_3_) in a dark environment for 1 h. In the following sections, the sensing films with different polymer matrices were labeled according to the blending ratio, i.e., SF-EC referred to the sensing film with EC as matrix and SF-E1P3 represented the film with the EC/PMMA ratio of 1:3.

### 3.3. Characterization of the Sensing Films

The absorption spectra of the experimental sensing films were measured in an UV-3600 UV-VIS-NIR spectrophotometer. The emission spectrum was measured in a FLS980 fluorescence spectrometer with an excitation wavelength of 535 nm. The surface of the sensing films was first sprayed with a thin layer of platinum to become conductive and then observed by field emission scanning electron microscopy (FE-SEM) of the type Gemini 500. The autofluorescence of the sensing film was photographed by the Nikon Eclipse Ti2-E Laser Confocal Microscope.

### 3.4. Experimental Setup and Measurement Principle

Figure 1a shows the schematic diagram of the fluorescence sensor used in this experiment. The excitation light, with a center wavelength of 535 nm, was emitted via a green LED and passed through the optical filter to be reflected by a dichroic mirror to the sensing film. Then, the sensing film was excited and emitted fluorescence that passed through the dichromatic mirror to reach the photodetector. In order to collect the original excitation light signal, the reference light (535 nm) with the same phase as the excitation light also reflected into the photodetector via the dichroic mirror at the same time. After a processing series by a specific algorithm, the phases of reference light (*θ*_r_) and fluorescence (*θ*_f_) could be acquired, as could the delay phase between them (Δ*θ* = *θ*_r_ − *θ*_f_, Figure 1b).

Based on the above principle, an instrument used for characterizing the performance of the sensing film is schematically shown in Figure 2. The sensing film was integrated with an excitation light source, dichromatic mirror, optical filters, and reference light source into an oxygen sensor. The whole system was then placed in a container that could withstand a pressure of up to 400 kPa. The pressure tank was sealed in the constant temperature trough and surrounded by deionized water to ensure a temperature accuracy of <0.1 °C.

Prior to the measurements, the tank was pumped to vacuum (−100 kPa vs. 1 atm), followed by inflating 100 kPa of nitrogen. The tank was then pumped twice to ensure an oxygen-free environment, and the lag phase of the sensing film in the anaerobic phase was recorded. Then, the delay phase of the experimental sensing film was obtained by inflating different concentrations of oxygen that ranged from 10% to 100%, which corresponded to the oxygen partial pressure of 10–100 kPa under normal atmospheric pressure. Oxygen partial pressures above 100 kPa were achieved by inflating oxygen with 100% concentration above normal atmospheric pressure. All data are averages of phase angles within 60 s.

## 4. Results

### 4.1. Surface Morphology of the Sensing Films

Since all the prepared sensing films were the same color, here, we only present the macro image of the sensing film with pure EC in Figure 3, where the sensing film is reddish on the surface and through the depth of the microporous membrane. Note that the microporous membrane without an indicator and polymer is white (see Appendix A), and all the prepared sensing films show reddish. Moreover, based on the autofluorescence images of the experimental sensing films (Figure 4), the autofluorescence of the indicator on each sensing film is evenly distributed on the polypropylene fibers of the filter membrane, and it could be inferred that the indicator has been fully absorbed by the microporous membrane.

The microstructure of the experimental sensing films was obtained by SEM (see Figure 5), and the diameter of the fibers falls in the range of 1–4 μm (Figure 5a). The polymer matrix in SF-EC, SF-E3P1, and SF-E1P1 formed as network structures within the polypropylene fibers, suggesting an unbalanced tension caused by local surface tension change during the volatilization of the solvent. However, when the PMMA content was increased in the polymer matrix, the network structure disappeared (Figure 5e,f). This can be attributed to the fact that the molecular mass of PMMA is greater than that of EC; hence, the viscosity of the configured solution increased with the increase of the specific gravity of PMMA in polymer blending. In addition, higher viscosity can impede the flow of the liquid polymer, producing a less obvious network structure during the evaporation of the solvent.

Based on the above observation, the adsorption method in this work can efficiently obtain sensing films with strong structural stability, and therefore, the polymer and indicator mixture on the microporous filter membrane fiber is stable enough to fall off.

### 4.2. Optical Properties of the Sensing Films

The absorption and emission spectra of the sensing films with different polymer matrices under ambient conditions (~25 °C and ~101 kPa) are further illustrated in Figure 6. The absorption peaks of the experimental sensing films are identical, located at 375 nm (Soret band), 505 nm, and 535 nm (Q band). This suggests that the performance measurement here can be achieved without changing the sensor hardware (i.e., LEDs and filters). Although the highest intensity of absorption peaks for the sensing film is 375 nm, this shorter wavelength of ultraviolet (UV) carries higher energy and accelerates the light bleaching effect, greatly shortening the indicator life [37]. Hence, the green light with a wavelength of 535 nm is used as the excitation light here. The emission peaks induced by the excitation light (535 nm in wavelength) of the experimental sensing films were 645 nm (Figure 6b); however, the fluorescence intensity of the sensing film with 100 wt.% EC is the lowest, and it increases with the increase of the PMMA content. This can be attributed to the fact that the oxygen permeability of PMMA is much lower than that of EC, and the different blending ratios of these two polymers can lead to different oxygen permeability of the sensing films.

### 4.3. Response Time of the Sensing Films

Figure 7 displays the phase changes for the experimental sensing films with different polymer matrix changes from air to vacuum, and then from vacuum to air, under ambient temperature in a pressure tank. In order to monitor the tank pressure in real time, a pressure gauge was placed within the tank. The pressure changes inside the tank, therefore, represent the change of the oxygen partial pressure. The phase changes of all the sensing films are almost synchronized with the change of the environmental pressure, suggesting a fast response of the experimental sensing films towards environmental oxygen. Since the sampling frequency of the applied monitoring system is 1 Hz, a time less than 1 s cannot be detected; the dynamic response time in this work is therefore considered as less than 1 s.

The response time (T_90up_ and T_90down_) of the sensing films with different polymer matrices is further tabulated in Table 1. Due to the fact that the pumping process is slower than the degassing process, T_90up_ values of the sensing films are generally higher than T_90down_. Therefore, the T_90down_ here is the criteria for evaluating the response performance of the experimental sensing films. From Table 1, the values of T_90down_ for all the samples are 3 s and 4 s; as the sampling frequency is 1 Hz, the response time of the sensing films with different polymer matrices can be treated as the same. In addition, the T_90up_ values of SF-E1P3 and SF-PMMA samples are slightly lower than those of other samples. This could be attributed to the fact that the viscosity of the PMMA solution is higher than the EC solution, leading to a more even coverage of mixture (indicator and polymer) around the fiber, and the mixture film on the fiber surface with a certain PMMA content (75 wt.%–100 wt.%) is relatively thinner (Figure 5e,f). Therefore, oxygen molecules are more likely to leave the film, and the phase of sensing films with lower-viscosity polymer substrate can reach a stable state faster during pumping.

### 4.4. Sensitivity and Dynamic Range of the Sensing Films

The phase change curves of the sensing films with different matrices under different oxygen partial pressures are shown in Figure 8a, where the detecting limits of the experimental sensing films are determined, as the phase change is less than 1° and the corresponding phase is not recorded in the curves. The phases of the sensing films are all decreased with the increase of *P*_O__2_, suggesting an obvious quenching effect of the indicators toward oxygen. The phase decreasing rate of the SF-EC sample is higher than that of other samples in the *P*_O__2_ range from 0 to 70 kPa, while the phase decreasing rate of the experimental sensing films is decreased as the PMMA content of the blending polymer matrix is increased from 25 wt.% to 100 wt.%. Oppositely, the detecting limit of the sensing films increased from 70 kPa to 180 kPa as the PMMA content in the blending polymer matrix increased from 0–100 wt.%, demonstrating an increased dynamic range of the sensing film with the increase of PMMA content.

Figure 8b shows the SVPs of the sensing films with different matrices at 20 °C, which are derived from Figure 8a. The slope of the curve, *K*_SV_, is defined as the sensitivity of the sensing film. The SVPs of the sensing films with different matrices all exhibit excellent linearity. Moreover, the *K*_SV_ value of the SF-EC sample is 0.298 kPa^−1^, which is higher than other samples. On the other hand, the *K*_SV_ values of the sensing films are decreased with the increase of PMMA content in the blended polymer matrix, reaching the lowest value of 0.017 kPa^−1^ when the polymer matrix is pure PMMA (Table 2). It is worth noting that the SVPs of sensing films with a blending polymer matrix of EC/PMMA ratios of 3/1 and 1/1 are not perfectly linear in a low-oxygen environment (*P*_O__2_ = 0–10 kPa). The reason could be ascribed to the different quenching degrees of indicator molecules in heterogeneous microenvironments. However, the SVPs of the sensing films with a blending polymer matrix still exhibit good linearity when the oxygen partial pressure is beyond 10 kPa, suggesting that they are promising for application over a broad range of oxygen concentrations.

### 4.5. Temperature Compensation of the Sensing Films

The temperature affects the fluorescence intensity of the indicator and thus the sensitivity of the sensing film [38,39,40]; however, most previous studies were carried out at room temperature. In order to investigate the effect of temperature on the sensing performance of the sensing films, we measured the phase delay variations of the sensing films with different types of the polymer matrices in the constant temperature trough (Figure 2), from 10 °C to 50 °C (with an increment of 10 °C). The oxygen partial pressure for each sensing film is based on the dynamic range shown in Table 2. Figure 9 presents the SVPs of the experimental sensing films from 10–50 °C, in which the phase delay of all the sensing films decreased as the temperature increased. This indicates that the quenching degree of the indicators increases with the environmental temperature. The underlying mechanisms can be attributed to the following aspects: (i) the rise of temperature aggravates the collision quenching between indicators and oxygen molecules [32]; (ii) the viscosity of the matrix is reduced with the increase of temperature, which provides more sites for the contact between indicators and oxygen molecules and promotes the quenching of the indicators in the polymer matrix. Moreover, the quenching values of tan *θ*_0_/tan *θ* for the sensing film with pure EC is higher than other samples at different temperatures, and the delay phase of the sensing films with the blending polymer matrix is increased with the decrease of the EC/PMMA ratio.

Since the collision quenching effect is determined by the diffusion of oxygen molecules, and the viscosity change of the polymer matrices with temperature is determined by their remelting point, the temperature sensitivity cross effect of the sensing film cannot be avoided, and it is necessary to deduce the temperature compensation function for each sensing film in practical application [38]. In this work, the temperature compensation for the sensing films is conducted by calibrating the sensing films in different temperatures (Figure 9), and the mathematical relationship between temperature and the quenching constant of the sensing films can be established by fitting the quadratic or linear coefficient of different types of the sensing films in different temperatures. Consequently, the temperature compensation function can be introduced to fix the S-V equations in a varying temperature environment.

Except for sample SF-E1P1, the quenching curves of sensing films are biased towards the *Y*-axis with the rise of temperature. One possible explanation is the hybrid quenching process of the sensing films, which includes both static quenching and dynamic quenching and can be defined using the following equation [41,42]:
(3)$${\tan\theta }_{0}/\tan\theta =1+(k_{d}+k_{s})P_{O2}+k_{d}k_{s}P_{O2}{}^{2}$$
where *k*_d_ is the dynamic quenching constant and *k*_s_ is the static quenching constant, namely the constant for the formation of the oxygen-indicator complex. In order to facilitate the calculation, we rewrite Equation (3) in a quadratic polynomial:(4)$${\tan\theta }_{0}/\tan\theta =AP_{O2}{}^{2}+BP_{O2}+C$$
where *A* and *B* are the quadratic and linear coefficients related to temperature, respectively, and *C* is a constant.

Appendix A summarizes the fitting parameters of the experimental sensing films based on Equation (4) at different temperatures. The *A* values of SF-EC and SF-E3P1 are nearly one order of magnitude higher than that of other samples. It is worth noting that the *A* values of SF-E1P1 are in the range of 8.22 × 10^−6^ − 1.81 × 10^−5^, showing good linearity for its S-V curve. We thus linearly fitted the S-V plot of the SF-E1P1 sample, and the results in Figure 9d exhibit excellent linearity (R^2^ > 0.99), indicating that the sensing films with a polymer matrix containing equal weight contents of EC and PMMA are more readily calibrated for practical use. In addition, *A* and *B* values for all the experimental sensing films can be defined as the quadratic functions of temperature (Figure 10). This is important for the application of the oxygen sensors, since equations for temperature compensation can be introduced by the fitting results for each type of the sensing films when calibrating the sensor in a temperature varying environment. It can be seen that the *A* and *B* values of the sensing films with the blending polymer matrix all show a good quadratic relationship in the temperature range from 10 °C to 50 °C, suggesting that the sensor matrices with the blending of different types of polymers are promising in applications where a broad temperature range is required. In addition, it is worth mentioning that the *B* value of SF-EC at 50 °C displays a non-quadratic function relationship with the previous temperature value; thus, it is not suitable for environments with a temperature of 50 °C.

## 5. Discussion

### 5.1. Sensitivity and Dynamic Range

It has been widely reported that the fluorescence change of nearly all the indicators, such as PtOEP, Ru(Ph_2_phen)_3_Cl_2,_ and Ru(dpp)_3_(ClO_4_)_2_ in the polymer matrix, are prone to be stable when the environmental oxygen content reaches a high value [43,44]. Therefore, it can be inferred that the polymer matrix would suppress the quenching between oxygen molecules and indicators, and the dynamic ranges of the sensing films would be modified by the amount of effective indicators that are in contact with oxygen molecules. Here, due to higher oxygen permeability, the indicators in the EC matrix can be in contact with more oxygen in a low-concentration oxygen environment when compared with the PMMA matrix. This results in a higher quenching efficiency as well as higher sensitivities for sensing films with a pure EC matrix. When the environmental oxygen content reaches a certain level (more than 70 kPa in this work), the indicator molecules in the EC matrix will be quenched completely or reach an equilibrium state, leading to no obvious phase change (1° in the current work) of such sensing film under higher oxygen pressure. At the same time, the sensing film with the PMMA matrix still possesses enough indicators that can be quenched by oxygen, and the phase change of sensing film with PMMA can be recognized when the environmental oxygen content continues to increase, resulting in a wider dynamic range. Moreover, the sensitivity and dynamic ranges of sensing films with the polymer matrix blended with different EC/PMMA ratios are within the range between sensing films with pure EC and PMMA. This is a clear indicator of the successful tuning effect via the physical blending of different types of polymers, which in principle modify the molecular gaps of polymers that host the fluorescence indicator. Most of the previous works have focused on the preparation of high-sensitive sensing films; compared with several earlier works listed in Table 3, in the case of similar sensitivity value, the sensing films with the polymer matrix of EC/PMMA ratios with 1/1 have the widest dynamic range of 10–150 kPa. In addition, compared to other sensing films with PMMA as the matrix, the sensing film in this work exhibits the largest dynamic range of 0–180 kPa.

### 5.2. Intercept of the Stern-Volmer Plot

According to Stern-Volmer equation (Equation (1)), in the absence of oxygen (i.e., *P*_O__2_ = 0 kPa in an ideal case), the tan *θ*_0_/tan *θ* value should be 1. That is, the SVP of the sensing films should intersect the (*x* = 0, *y* = 1) point on the graph. In the present work, the intercepts of SVPs of SFs with pure EC and PMMA matrices and the matrix of EC/PMMA with a ratio of 1/3 under different temperatures are close to 1. However, the intercepts of the SVP for the sensing films with a blending polymer matrix of different EC/PMMA ratios with 3/1 and 1/1 show a deviation from 1. The underlying mechanisms are unfortunately not clear yet. It is worth noting that Lee et al. [35] fabricated oxygen-sensing film with PtOEP embedded in polystyrene (PS) polymer matrix. According to their experimental results, the slopes of the sensing films’ SVP are larger than 1 when oxygen concentration (O_2_) is higher than 15%, while the intercepts for the sensing films’ SVPs in the (O_2_) range from 0 to 15% are close to 1. Due to the limitation of our instruments, the SVPs for the present sensing films in the *P*_O2_ range from 0–10 kPa cannot be achieved. It therefore can be inferred that because the slopes of SVPs for the present sensing films with blended polymer matrices might be changed when *P*_O2_ is less than 10 kPa and the intercepts of their SVPs are not equal to 1 when *P*_O2_ is larger than 10 kPa.

## 6. Conclusions

In this study, a simple and low-cost approach for tuning the sensitivity and dynamic range of optical oxygen sensing film is achieved by the physical blending of EC and PMMA with different mixing ratios as sensing film matrices. The use of a polypropylene microporous filtration membrane as the support allows the uniform distribution of indicators on the sensing film surface. This structure endows the sensing films with a fast dynamic response time, which is less than 1 s.

The sensitivities and dynamic ranges of the sensing films are mainly determined by the oxygen permeability of the polymer matrix. The sensing film with pure EC, which possesses the highest oxygen permeability, exhibits the highest sensitivity of 0.298 kPa^−1^ and smallest dynamic range of 0–70 kPa, while the sensing film with the PMMA matrix with the lowest oxygen permeability exhibits the lowest sensitivity of 0.017 kPa^−1^ and widest dynamic range of 0–180 kPa. The sensitivities and dynamic ranges of the sensing films with the blended polymer matrix are decreased and increased as the EC/PMMA ratio decreased, respectively, achieving a good balance that is favorable for the application over a broad range of oxygen concentrations.

In addition, the S-V curves of the sensing film with equal EC and PMMA contents show good linearity at different temperatures, which is more readily calibrated for temperature varying environments.

## Figures and Tables

**Figure 1 biosensors-12-00005-f001:**
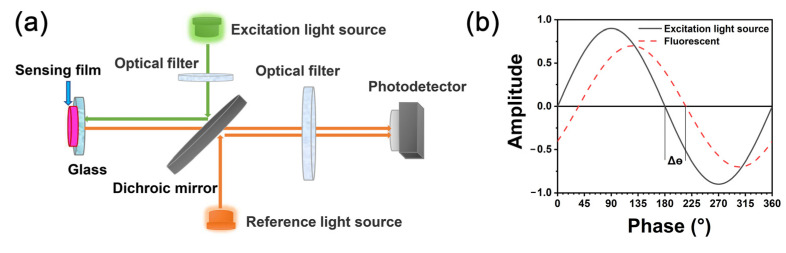
(**a**) Schematic diagram of fluorescence sensor; (**b**) principle of phase delay method.

**Figure 2 biosensors-12-00005-f002:**
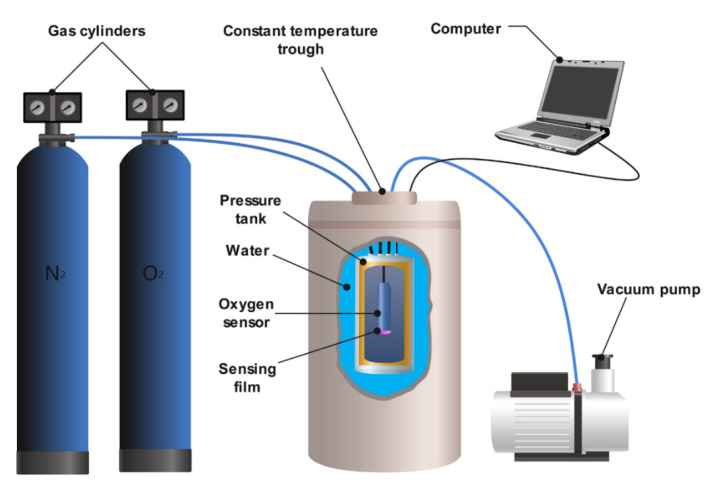
Schematic diagram of the experimental apparatus used to test the performance of the sensing film.

**Figure 3 biosensors-12-00005-f003:**
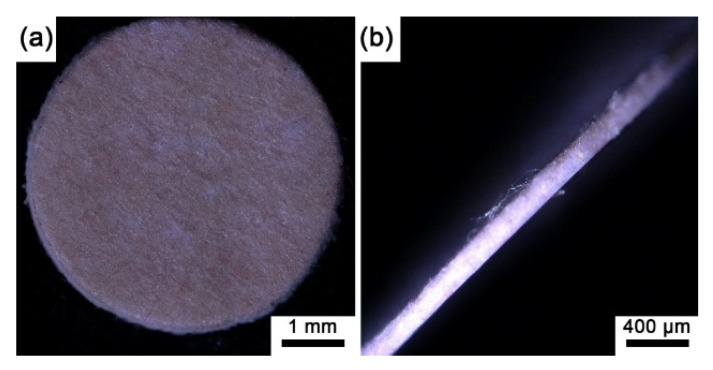
(**a**) The optical image of the sensing film (SF-EC) with a diameter of 5 mm; (**b**) side view of the sensing film (SF-EC) with a thickness of 160 μm.

**Figure 4 biosensors-12-00005-f004:**
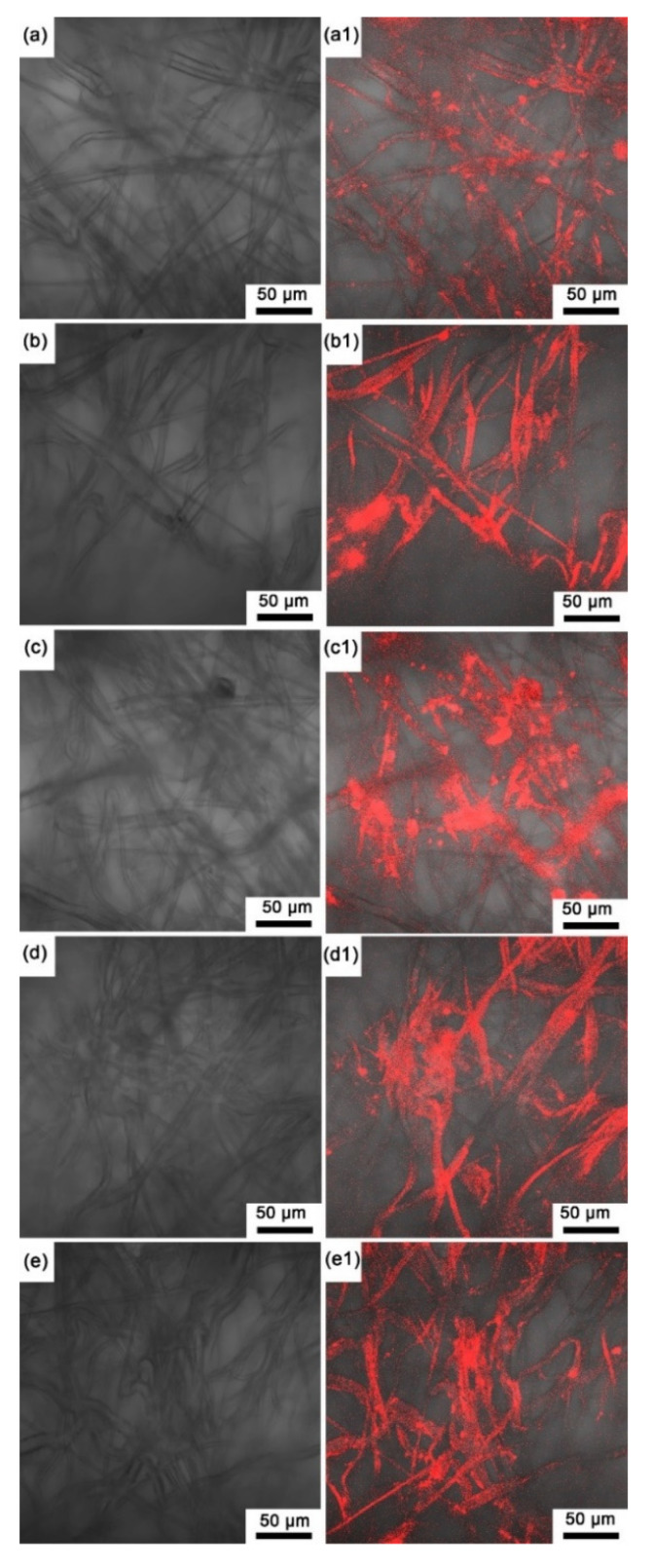
Unexcited images of the sensing films (**a**) SF-EC; (**b**) SF-E3P1 (EC/PMMA ratio of 3:1); (**c**) SF-E1P1 (EC/PMMA ratio of 1:1); (**d**) SF-E1P3 (EC/PMMA ratio of 1:3); (**e**) SF-PMMA. Autofluorescence images of the sensing films (**a1**–**e1**).

**Figure 5 biosensors-12-00005-f005:**
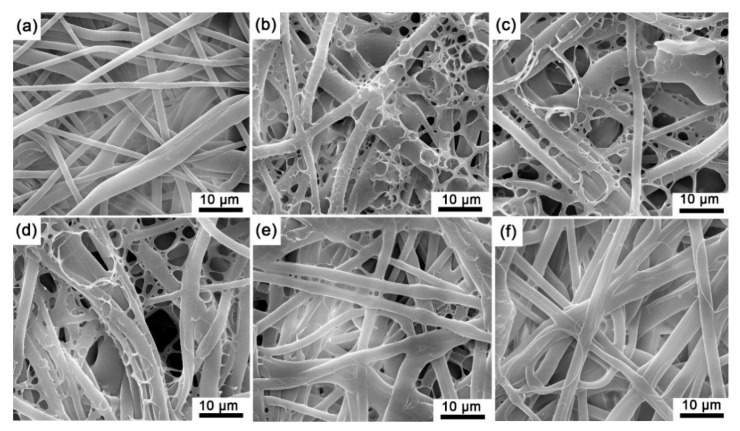
SEM images of (**a**) polypropylene microporous membrane; (**b**) SF-EC; (**c**) SF-E3P1 (EC/PMMA ratio of 3:1); (**d**) SF-E1P1 (EC/PMMA ratio of 1:1); (**e**) SF-E1P3 (EC/PMMA ratio of 1:3); (**f**) SF-PMMA.

**Figure 6 biosensors-12-00005-f006:**
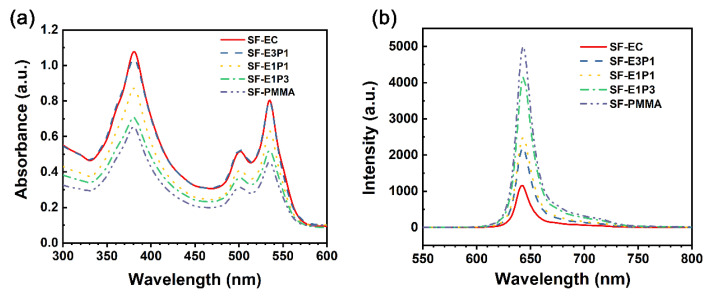
(**a**) Absorption spectra and (**b**) emission spectra of the sensing films with different polymer matrices under ambient conditions.

**Figure 7 biosensors-12-00005-f007:**
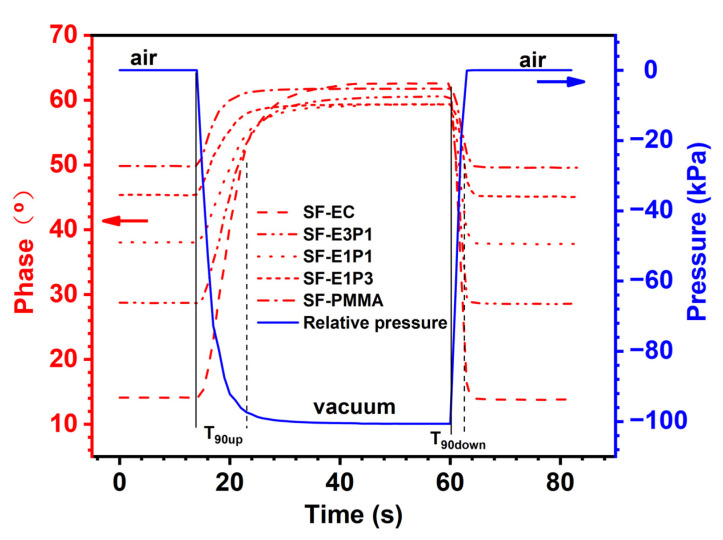
Phase changes of the sensing films with different polymer matrices from air (0 kPa) to vacuum (−100 kPa), and from vacuum to air, under ambient temperature in a pressure tank.

**Figure 8 biosensors-12-00005-f008:**
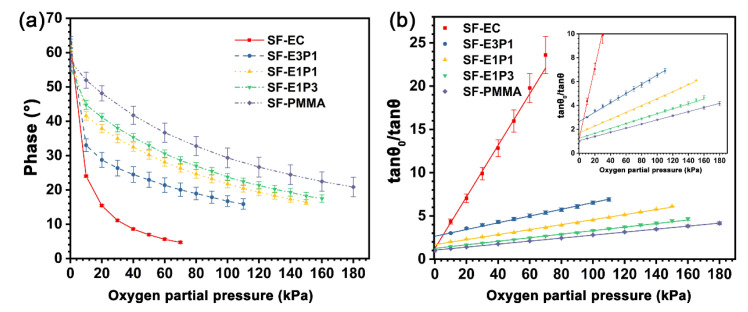
(**a**) Phase change of the sensing films with different matrices under different oxygen partial pressure at 20 °C; (**b**) Stern-Volmer plots of the sensing films derived from (**a**). Inset: local amplification.

**Figure 9 biosensors-12-00005-f009:**
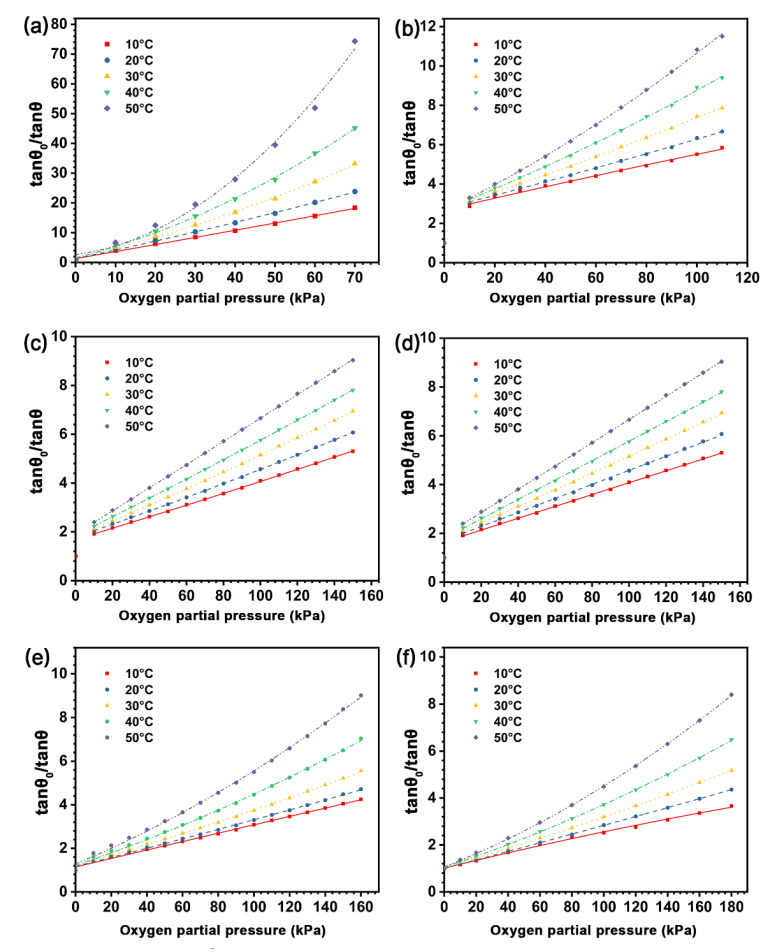
Stern-Volmer plots of the sensing films with different polymer matrices of (**a**) SF-EC; (**b**) SF-E3P1 (EC/PMMA ratio of 3:1); (**c**) SF-E1P1 (EC/PMMA ratio of 1:1); (**d**) linear fitting of SF-E1P1; (**e**) SF-E1P3 (EC/PMMA ratio of 1:3); (**f**) SF-PMMA from 10 °C to 50 °C. Except for (**d**), fitting formulas are quadratic polynomials and R^2^ > 0.99.

**Figure 10 biosensors-12-00005-f010:**
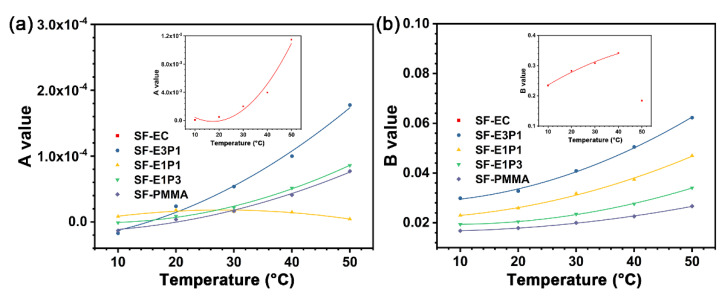
(**a**) *A* value and (**b**) *B* value fitted with a quadratic polynomial base varies with temperature. The fitting formula is a quadratic polynomial, and R^2^ > 0.95. Inset: *A* value (**a**) and *B* value (**b**) of SF-EC.

**Table 1 biosensors-12-00005-t001:** Response time of the sensing films with different polymer matrices.

Sample	Dynamic Response time ^a^ (s)	T_90up_ ^b^/ Air-Vacuum (s)	T_90down_ ^c^/ Vacuum-Air (s)
SF-EC	<1	13	3
SF-E3P1	<1	13	3
SF-E1P1	<1	13	3
SF-E1P3	<1	10	4
SF-PMMA	<1	8	4

^a^ Dynamic response time: the time difference between the pressure changes and the phase change. ^b^ T_90up_: the time for 90% of the overall phase increase from 0 kPa to −100 kPa. ^c^ T_90down_: the time for 90% of the overall phase decrease from −100 kPa to 0 kPa.

**Table 2 biosensors-12-00005-t002:** Performance characteristics of sensing films with a different matrix.

Sample	*K*_SV_^a^ (kPa^−1^)	Dynamic Range (kPa)
SF-EC	0.298	0–70
SF-E1P3	0.039	10–110
SF-E1P1	0.029	10–150
SF-E3P1	0.021	0–160
SF-PMMA	0.017	0–180

^a^*K*_SV_: the slope of the Stern-Volmer curve (sensitivity).

**Table 3 biosensors-12-00005-t003:** Comparison of performance characteristics of the current sensing film with existing sensing films.

Indicator	Polymer Matrix	Sensitivity/*K*_SV_ (kPa^−1^)	Dynamic Range (kPa)	Ref.
PtOEP	EC	0.298	0–70	This work
PtOEP	EC/PMMA ratio of 3:1	0.039	10–110	This work
PtOEP	EC/PMMA ratio of 1:1	0.029	10–150	This work
PtOEP	EC/PMMA ratio of 1:3	0.021	0–160	This work
PtOEP	PMMA	0.017	0–180	This work
PtTPTBPF_4_	tButPS	0.477	0–55	[31]
PtOEP	poly(p-FSt-co-TFEMA)	0.180	0–100	[2]
PtOEP	PMMA	0.022	0–100	[19]
PtOEP	PS/PEG ratio of 4:1	0.129	0–100	[35]
PtOEP	PS/PEG ratio of 9:1	0.153	0–100	[35]
Ru(bpy)_3_]^2+^Cl_2_·6H_2_O	PMMA	0.0026	0–100	[45]
Ru(bpy)_3_]^2+^Cl_2_·6H_2_O	PSS	0.0230	0–100	[46]

## Data Availability

The raw data required to reproduce these findings are available upon request from the corresponding author (Siyuan Lu).

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
