# Peer review of "Tuning the Sensitivity and Dynamic Range of Optical Oxygen Sensing Films by Blending Various Polymer Matrices"

_biosensors, 2021, doi:10.3390/bios12010005_

Round 1

Reviewer 1 Report

In this work, the authors prepared different types of oxygen sensing films and characterized their morphology, fluorescence spectra. The content dependent sensitivity and dynamic range have studied and compared. Some comments are suggested to further improve this manuscript:

  1. Some details should be pay attentions: “sensing film” is not the special or a too long words, you do not need to explain them using “SF”. Furthermore, this word also turns out in the manuscript in its full name. They must be unified; Page 2, Line 47, PMDS should be PDMS; Page 4, Line 161, signa, should be signal; Page 16, line 415, Grant No. 11,702,157 should be 11702157…. etc.
  2. Some figures and tables should be rearranged and placed in the proper locations through the main text of the manuscript.
  3. “the” should be deleted from the section titles;
  4. The impact of temperature on the sensing performance seems too high by comparing Figure 8 and Figure 9. This impact and the possible method for reducing the temperature sensitivity-cross effect should be discussed in detail.
  5. The potential application and advantages of this work should be explained. The possible impact from water and temperature should be discussed for the actual application.
  6. The variable symbols on the coordinate axis in the figures are suggested to be replaced into the text form, such as Phase change (θ), Oxygen partial pressure (PO2), etc…

Reviewer 2 Report

The authors demonstrated tuning the sensitivity and dynamic range of optical oxygen sensing films by blending various polymer matrices. The method is novel. Through this cost-effective method, the sensitivity and dynamic range of 82 optical oxygen sensing films can be adjusted. Overall the paper is well written and good organized, but related certain points must be explained in more details. Following modifications are required:

  1.  In the part of introduction, it is recommended to show the corresponding sensitivity values and dynamic range obtained by other methods for comparison with the method in this article.
  2. In the introduction, “Moreover, tunning the sensitivity by altering the polymer types also encounters great intervention for a specific sensor system, since the matrix of SF only can be used  in a specific environment”. Therefore, What are the scenarios in which the method in this article can be applied? It is recommended to elaborate.
  3.  In the summary part, it is almost the same as the conclusion of the introduction part. It is recommended to display the corresponding sensitivity value, the size of the dynamic range and the corresponding response time or other important information of this method to summarize the effect.

Reviewer 3 Report

The work presented in this manuscript is novel, interesting, and well-suited for publication in biosensors. The authors study the performance of different optical oxygen sensing films prepared from impregnating PtOEP with different blends of polymer matrices (EC/PMMA mixtures). There are just a few minor points which need to be addressed before this publication can be accepted.

  1. Please also either indicate in the figure captions or in the results and discussion, the abbreviations for E131, E1P1, E3P1, to improve readability and understanding. It takes a while to understand the abbreviation refers to different ratios of EC to PMMA because it is only mentioned in the experimental section.
  2. Please also indicate in Table 2 that Ksv refers to the slope of the curve (sensitivity).
  3. In section 5, the discussion should also include a more quantitative discussion of how the authors’ oxygen sensor compares to other types of oxygen sensor in terms of sensitivity and range. Currently, there is almost no comparison to other oxygen sensors.
  4. For example, electrochemical detection techniques are commonly combined with conjugated polymers such as polythiophene, polyaniline, or polypyrrole. How does the authors’ sensor compare to such sensors? Please expand on the discussion in the manuscript, and here is a review you can consider citing (http://dx.doi.org/10.1016/B978-0-12-803581-8.10144-4).

Round 2

Reviewer 1 Report

It can be accepted as its present form

Reviewer 2 Report

The revised manuscript has been well modified. It can be accepted.